
**Levee breach-induced compound flood modeling in Qianbujing Creek, Shanghai**
**during Typhoon "Fitow"**
Yuhan Yang[1], Jie Yin[2, 3, 4*], Weiguo Zhang[1], Yan Zhang[2], Yi Lu[2], Aoyue Xiao[2], Yunxiao Wang[2],
Wenming Song[2]
1 State Key Laboratory of Estuarine and Coastal Research, East China Normal University, China
2 Key Laboratory of Geographic Information Science (Ministry of Education), East China Normal
University, China
3 Department of Civil and Environmental Engineering, Princeton University, USA
4 Institute of Eco-Chongming, East China Normal University, China
* Correspondence to: J.Y. (email: jyin@geo.ecnu.edu.cn)
*Competing interests*. The authors declare that they have no conflict of interest.
**Abstract:** Levee breach-induced flooding occurs occasionally but always causes considerable
losses. A serious flood event occurred due to the collapse of a 15-m-long levee section in
Qianbujing Creek, Shanghai, China, during typhoon "Fitow" in Oct, 2013. Heavy rainfall
associated with the typhoon intensified the flood severity (extent and depth). This study
investigates the flood evolution to understand the dynamic nature of flooding and the compound
effect using a well-established 2D hydro-inundation model (Floodmap) to reconstruct this
typical event. Our simulation results provide a comprehensive view of the spatial patterns of





the flood evolution. The worst-hit areas are predicted to be low-lying settlement and farmland.
Temporal evaluations suggest that the most critical time for flooding prevention is in the early
hours after dike failure. In low-elevation areas, temporary drainage measures and flood
defenses are equally important. The validation of the model demonstrates the reliability of the
approach.
**Key words:** levee breach; compound flooding; inundation modeling; Shanghai



















**1.   Introduction**

Flooding is a common and devastating natural hazard, causing considerable personal injury,
loss of life, and property damage worldwide (Jonkman et al., 2005; Jongman et al., 2012).
Engineering measures such as dikes and barriers are typically constructed in low-lying deltas
and floodplains to prevent flooding. However, weak or aging dikes without regular maintenance
may fail during extreme flood events. Levee breaches may result in extensive flooding and
damages throughout the hinterland (Ying et al., 2003). For example, Hurricane Katrina-induced
flooding significantly damaged the dike system of New Orleans and overwhelmed the city,
making it the costliest disaster in U.S. history (Kates et al., 2006). A more recent flood
catastrophe with more than 50 deaths and hundreds of missing people resulted from of a dam
breach due to a Himalayan glacier outburst flood in northern India.

In addition, riverine or storm-induced flooding is typically associated with heavy precipitation.
The compound effect of pluvial, fluvial, or coastal flooding is much greater than the effect of
individual flood events (Wahl et al., 2015). For instance, typhoon "Fitow" in 2013 brought
torrential rain and caused high storm surge, resulting in record-breaking riverine water levels
in the upstream region of the Huangpu River, Shanghai, China. As a result, the floodwall along
the upstream Qianbujing Creek could not withstand the high water level, leading to a breach in
a 15-m long section at 14:30 on 8 Oct 2013. Although the broken section was repaired after
about 8 hours, the levee breach combined with heavy precipitation resulted in extensive flood
inundation in the rural areas.




Over the last few decades, dike failure-induced flooding and the compound effect have received
increasing attention from decision-makers, researchers and even general public. Recent studies
have provided considerable progresses on dike reliability analysis and compound flood
modeling (Curran et al., 2018; Naulin et al., 2018). A number of approaches for levee breach-
induced flood modeling were developed. For example, Vorogushyn (2010) proposed an
Inundation Hazard Assessment Model (IHAM), which coupled a 1D hydrodynamic model of
river channel routing, a probabilistic dike breach model, and a 2D raster-based inundation
model. Cannata et al. (2011) used a GIS-based approach to simplify a 2D dam break simulation.
Yin et al. (2020) predicted dike failures and flood inundations in Shanghai, China, under various
emission scenarios using an interdisciplinary process-based approach.

Similarly, numerous studies analyzed the compound effects of various flood hazards at different
scales (Ganguli et al., 2020). Lian et al (2013) evaluated the combined effect of rainfall and the
tidal level on flood risk in a complex river network in a coastal city in China. Moftakhari et al
(2017) proposed a bivariate flood hazard assessment approach to account for compound
flooding from river flow and coastal water level. Bevacqua et al (2019) predicted the increasing
probability of compound flooding from precipitation and storm surges in Europe under
anthropogenic climate change. At a global scale, Couasnon (2020) and Eilander (2020)
explored the compound flood potential resulting from storm surges and riverine floods.

The above studies contributed significantly to the modeling or evaluation of dike failure-



induced flooding or compound flood risk. However, few studies investigated the compound
influences of pluvial and levee breach-induced flooding or focused on the dynamic process and
mechanism of these cases, what can guide the development of appropriate emergency response
plans. Moreover, real-life cases of historical flooding events have not been adequately
investigated but can demonstrate the feasibility and robustness of the model. To address the
research gaps, this case study seeks to examine the changing nature of levee breach-induced
compound flooding. A simple 2D hydro-inundation model Floodmap is used to simulate the
process of the compound flood event that occurred in Qianbujing Creek to improve our
understanding of the evolution of flood inundation. The results of the approach are validated
by field measurements, including the inundation depth and the flood extent over time. The
findings can provide support for decision-makers to develop flood adaptation measures.

**2.   Materials and methods**

**2.1 Study area**

The study area is located at the junction of the Huangpu River and Qianbujing Creek in the
upstream Huangpu River Basin, Shanghai, China. The rural area covers about 1.5 km$^2$ with
majority of agricultural land and minority of human settlements. It is characterized with a mild
and low-lying topography (with an average altitude of about 3 m above Wusong Datum). Due
to its location, the study area has faced high flood risk from the river system; however, the
heights of the flood defense measures are relatively low (i.e., a 50-year return period flood





protection standard) compared to the high floodwall (1000-year period flood protection
standard) along the mid- and downstream urban regions of the Huangpu River (Yin et al., 2020).
Furthermore, because the region has a northern subtropical monsoon climate, pluvial flood
events caused by extreme rainfall, typically associated with typhoons, are frequently recorded
during the flood season (June to September) (Yin & Zhang, 2015). Therefore, the risk of
compound flooding from both riverine and pluvial sources is significantly higher than that in
other locations. Figure 1 shows the location of the study area and the levee breach during
typhoon "Fitow".

**2.2 Data sources and processing**

**2.2.1    Topographic data**

We used a 6-m resolution digital surface model (DSM) of the study area constructed from
images of the China Resource 3 satellite (ZY-3) and other high-resolution satellites. Since
buildings and trees represent barriers to water flow and reduce the area available for water
storage in the hydrodynamic model, we removed the non-topographic features (e.g., trees and
buildings) according to the Google historical dataset of remote sensing images to generate a
bare-earth digital elevation model (DEM) based on the Wusong Datum of Shanghai. (Fewtrell
et al., 2010; Neal et al., 2011; Yu & Lane, 2006b). We further resampled the cell size of the
bare-earth DEM from 6 m to 2 m using ArcGIS software to improve the spatial resolution of
the flood inundation model. The simulation domain of the study area consisted of 0.3 million



cells with an area of nearly 1.26 km$^2$.

**2.2.2     Precipitation and water level**

Time series of the precipitation and water level records during Typhoon "Fitow" were used as
boundary conditions to simulate the hydrodynamic process of the levee breach-induced
flooding and the rainfall runoff. The data are typically derived from the stage measurements at
gauge stations or radar-based rainfall data. However, due to the small scale of the study area,
the gauging records are considered to be more reliable. Thus, we collected the historical records
of the precipitation and water level data at the nearest gauging station from 0:00 on 8 Oct to
12:00 on 9 Oct 2013 for about 12 hours before and after the levee failure.

The station-based precipitation records (at one-hour intervals) were obtained from the
Information Center of the Shanghai Meteorological Administration. The water level data (at 5
min intervals) at the closest gauging station along the Huangpu River (i.e. Songpu Bridge
gauging station at the upstream of the Huangpu River, about 4 km away from Qianbujing Creek )
were provided by the Shanghai Municipal Water Administration. The time series of the rainfall
and water level data interpolated from the gauging stations is shown in Figure 2. Heavy rainfall
occurred four hours before the levee breach, with the maximum hourly rainfall exceeding 20
mm/h, resulting in the high water level of the river. Due to the high rainfall and rising storm
tide, the water level increased rapidly to nearly 4.8 m and caused the collapse of a 15-m-long
floodwall section at about 14:30 on 8 Oct.




### 2.2.3 Validation data


Aerial images or field surveys of flood extent were not available for the study event. There was
also a lack of water depth data from electronic gauges and flood incidents reported by the public.
Therefore, we validated the model through the field investigation of high water marks in the
study area. We visited the study area three times in 2020 and investigated the residential areas
(house by house), roadways, and farmland mostly affected by the flood event. Validation data
were collected using questionnaires, and the coordinates of the locations were recorded by GPS.
However, since this flood event occurred more than 7 years ago, there are inherent uncertainties
in the investigation due to the changing environment and people's fading memory for the details
of the event. Similarly, people tend to exaggerate their injuries and losses during hazards; thus,
questionnaires can be highly biased. Finally, we pinpointed 32 incidents in total where locations
are confidently identified. Among the 32 observed inundation data, 14 were buildings, 10 were
roadways, and 8 were farmland locations (Figure 3).

### 2.3 Levee breach modeling


In general, levee breach mechanisms mainly include structural instability failures and structural
strength failures. The former mode includes horizontal instability and rotational instability,
whereas the latter refers to the destruction of structures (Wang, 2016). Due to the configuration
of the floodwalls and the soil structure in Shanghai, structural instability failures always occur



during low water levels when critical inundation is less likely. In this case, structural strength
failure was considered the major reason for the levee breach in the study area, namely, the levee
collapse under an excessive hydraulic load on the wall due to an extremely high water level or
the uneven settlement of the floodwalls.

We identified the location of the levee breach from the historical news reports and through field
investigation. The 15-m long levee breach was located at the junction of Qianbujing creek and
the main channel of the Huangpu River. The remaining floodwall height (4.9 to 5 m above
Wusong Datum) data were obtained from the Shanghai Municipal Institute of Surveying and
Mapping. The data were then overlaid onto the original bare-earth DEM using the raster
calculator in ArcGIS 10.6 software.

**2.4 Compound flood modeling**

The compound flood modeling was performed using a 2D hydro-inundation model (FloodMap)
(Yu & Coulthard, 2015; Yu & Lane, 2006a; 2006b), which couples hydrological processes (e.g.,
infiltration, evapotranspiration, and drainage) module with 2D surface flood inundation
modeling. The Floodmap model provides an effective approach for compound flood simulation,
allowing for more than one hydrological boundary condition, including pluvial, fluvial, coastal
and groundwater sources. In this study, the compound effect of pluvial and fluvial flooding was
investigated. The fluvial flood modeling and pluvial flood modeling are described in Sections
2.4.1 and 2.4.2, respectively.




### 2.4.1 Fluvial flood modeling


For simulating the levee breach-induced flooding, a simplified flood inundation module based

on a raster environment was used to solve the inertial form of the 2D shallow water equations.

The module considered the mass and momentum exchange between the river flow and

floodplain inundation and has been used to simulate the dynamic nature of flood routing and to

extract potential flood maps (Yang et al., 2020; Yin et al., 2015). The 2D inundation model is

similar to the inertial algorithm of Bates et al. (2010). The difference is the time-step calculation

approach. The optimal time step is calculated using the subsequent iteration instead of using

the time step of the next iteration calculated by the current iteration. The main structure of the

model is presented below.

211

The Saint-Venant momentum equation without the convective acceleration has the following

form:

$$\frac{\partial q}{\partial t} + \frac{gh\partial(h+z)}{\partial x} + \frac{gn^2q^2}{R^{4/3}h} = 0 \tag{1}$$

where $g$ is the acceleration of gravity, $q$ is the flow per unit width, $R$ is the hydraulic radius,

$h$ is the water depth, $z$ is the bed elevation, and $n$ is Manning's roughness coefficient. For

wide and shallow flows, $R$ can be approximated with $h$. The equation discretized with respect

to time is:

$$\frac{q_{t+\Delta t}-q_t}{\Delta t} + \frac{gh_t\partial(h+z)}{\partial x} + \frac{gn^2q_t^2}{h_t^{7/3}} = 0 \tag{2}$$

The $q_t$ in the friction term can be replaced by $q_{t+\Delta t}$ to obtain the explicit expression in the





next time step:

$$q_{t+\Delta t} = \frac{q_t - gh_t\Delta t(\frac{\partial(h_t+z)}{\partial x})}{(1+gh_t\Delta t n^2 q_t/h_t^{10/3})} \qquad (3)$$

The flows in the $x$- and $y$-directions are decoupled and have the same form. The discharge is
evaluated at the cell edges, and the depth is determined at the cell center. For model constancy
and minimizing numerical diffusion, we use the forward Courant-Friedrichs-Lewy condition
(FCFL), which was used by Yu & Lane (2011) for the diffusion-based version of FloodMap, to
calculate the time step in the inertial model:

$$\Delta t \le \min\left(\frac{w d_i d_j n}{d_i^{1.67}(S_i)^{1/2} + d_j^{1.67}(S_j)^{1/2}}\right) \qquad (4)$$

where $w$ represents the cell size, $i$ and $j$ are the indices for the flow direction in the $x$- and
$y$-directions, $d_i$ and $d_j$ are the effective water depths; $S_i$ and $S_j$ are the water surface
slopes. The effective water depth is defined as the difference between the high water surface
elevation and the high bed elevation of two cells that exchange water. The minimum time step
that satisfies the FCFL condition for all wet cells is used as the global time step for this iteration.
This approach does not require the back-calculation of the Courant number because the time
step is calculated based on the CFL condition that satisfies every wet grid cell for the current
iteration. The universal time step calculated with the FCFL may need to be scaled further by a
coefficient with a value between 0 and 1 because the FCFL condition is not strictly the right
stability criteria for an inertial system. A scaling factor in the range of 0.5–0.8 was found to
yield a stable solution in previous studies; here, a scaling factor of 0.7 was used for all
simulations. The calibration and validation of the model for the study area were conducted by
Yin et al. (2016).

**2.4.2    Pluvial flood modeling**

In terms of the pluvial flooding module, we ran the surface water flood routing using the same
structure as the fluvial flooding module. The infiltration over saturation was calculated by the
widely used Green-Ampt equation, and the evapotranspiration was represented using a simple
seasonal sine curve of daily potential evapotranspiration (Calder et al., 1983). This module also
considered the amount of runoff loss to the urban storm sewer systems by scaling the drainage
capacity (mm/h) for each time step.

The infiltration over saturation was determined by the widely used Green–Ampt equation,
which approximates the rate of infiltration as a function of the capillary potential, porosity,
hydraulic conductivity, and time using the following form:

$$f(t) = K_s \left( \frac{\varphi_f + h_o}{z_f} + 1 \right) \tag{5}$$

where $K_s$ expresses the hydraulic conductivity of the saturated soil, $\varphi_f$ is the capillary
potential across the wetting front, $h_o$ is the water ponding on the soil surface, and $z_f$ is the
cumulative depth of infiltration.

The evapotranspiration was determined using a simple seasonal sine curve of daily potential
evapotranspiration (Calder et al., 1983) as follows:

$$E_p = \overline{E_p} \left[ 1 + \sin\left( \frac{360i}{365} - 90 \right) \right] \tag{6}$$

where $E_p$ is the mean daily potential evapotranspiration, and $i$ is the day of the year. The
mass lost to evapotranspiration is typically limited due to the short duration of urban pluvial



flooding.

We used the topography boundary conditions, flow boundary conditions, and precipitation
boundary conditions as inputs to model a 36-h compound flood process, including the 12 h
before and after the levee breach, and we assumed evapotranspiration of 3 mm/day. The soil
hydraulic conductivity ($K_s$) is an important but highly complex parameter used to calculate
infiltration. Empirically-based correlation methods or in situ hydraulic laboratory
measurements can be used to determine the value of $K_s$. Given the practical constraints, this
study refers to previous flood simulations in Shanghai (Yin et al., 2016; Yin et al., 2015; Yu &
Coulthard, 2015; Yu et al., 2016) and used the value of 0.001 m/h for the hydraulic conductivity.
A relatively high roughness value ($n$ = 0.06) was used in the simulation, according to the type
of cultivated land and crops in the study area. Since the Qianbujing creek is located in a rural
area, we did not consider the urban storm drainage capacity in this simulation.

**3.   Results**
**3.1 Time series of flood inundation**

Figure 4 shows the changes in the predicted flood inundation every 4 h during the event, and
Figure 5 depicts time series of average water depth and flood extent. These results show the
spatial and temporal evolution of the levee breach-induced compound flooding during typhoon
"Fitow". Prior to the levee breach, it is apparent from Figures 4 and 5 that heavy rainfall in the
study area led to localized shallow waterlogging, mainly in the low-lying farmland and forests.



The inundation area reached its first peak in the early hours on 8 Oct, but the water retention
time was very short due to the shallow water depth (< 15 cm). At around 11:00 am on 8 Oct,
another short-term rainstorm with rainfall over 20 mm/h occurred. Shortly after the
precipitation peak, the water level of Qianbujing Creek showed an increasing trend. The
compound effects of tide rising and heavy rain made the water level soon reached nearly 4.8 m
(Figure 2). Due to the high water pressure, the bearing capacity of the floodwall was exceeded,
resulting in a 15-m breached section (at 14:30 pm). Subsequently, overland flow through
breached floodwalls and extensive flood inundation occurred quickly along the riverbank, first
in the low-lying farmland near the river and then on roads and residential areas. About 10 homes
were completely inundated during the water level rising period (until 16:00 p.m.) with the
maximum inundation depth higher than 2 m. After 16:00 p.m., as the rainfall stopped and the
water level dropped, the inundation area gradually stopped spreading.

A cross comparison of the derived flood hazard maps over time further indicated that although
the rainstorm caused extensive surface water flooding in in majority of the study area, the
inundation depth was generally shallow (< 15 cm). This effect can be attributed to the
evapotranspiration and infiltration in a few hours. However, unlike the short-term waterlogging
caused by the rainstorm, the compound effects of the rainfall and levee breach-induced flood
inundation continued over 12 h, with an average water depth of nearly 60 cm.

**3.2 Maximum flood inundation**



The maximum flood extent and inundation during the event is shown in Figure 6. We use 2 cm
as the threshold for surface water flooding and treat water depths shallower than 2 cm as sheet
flow, which did not accumulate in topographic lows (Yu et al., 2016). Figure 6 shows that over
half (56%) of the study area inundated from the compound flooding, and most of the flooded
areas were low-lying farmland with maximum flood depths of higher than 2 m. Aside from the
waterfront areas, many low-lying farmland areas were affected by the rainstorm, with
maximum water depths over 50 cm. In contrast, the water depth on the roads and the buildings
was shallow; most of it was less than 0.5 m and disappeared quickly. In nearly half of the
flooded locations, the water depths were between 2 cm and 15 cm (44.1%), and a smaller
proportion of the area (21.12%) had water depths between 15 cm to 50 cm. About 33.26% of
the inundated areas had water depths of 50 cm to 2 m. In combination with the time series of
water level and rainfall (Figure 2), It can be inferred that the maximum flood inundation
occurred at about the fourth hour after the levee breaching (at ~16:00 p.m.) in waterfront area,
while it occurred at about 11:30 a.m. in other areas.

**3.3 Model validation**

The field measurements were used to validate the performance of the compound flooding model.
Figure 6 shows the location of the measurement points. The points were divided into building,
road, and farmland types. Since there are few residential areas in the study area, reliable
inundation information could not be obtained in most flooding areas; therefore, most of the
points represent buildings with extensive inundation. Since there were uncertainties and errors


---

in the survey results, including the respondents' memory bias, exaggeration of inundation, and
false positives, we set the observed error to 5 cm for building points, 10 cm for road points, and
15 cm for farmland points. The simulation error was set as 5 cm. Figure 7 shows the scatter plot
of the simulated and observed water depth and the 95% confidence interval. A correlation was
observed between the simulated water depth and observed water depth, and most points fell
within the confidence band. The observed water depth was slightly higher than the simulated
water depth, which may be attributed to the exaggeration of the water depth by the respondents.

**3.4 Sensitivity analysis**

The model sensitivity to Manning's roughness coefficient over time was analyzed. Several
Manning's n values (0.01–0.1 at a 0.01 increment) were used for the roughness
parameterization. The difference between the average water depth (Figure 8a) and the total
inundation area (Figure 8b) predicted by the simulations with different n values was calculated
on a cell-by-cell basis. The results indicate similar trends of the average water depth and
inundation area for different roughness values and differences in the values. As the roughness
increased, the average water depth decreased, and the difference was more pronounced at higher
roughness values. For example, the maximum average depth decreased from 0.55 m to 0.61 m
with an increase in the n value of 0.01 to 0.1. Interestingly, there were differences in the
sensitivity to the roughness before and after the levee breach for the flood inundation extent.
The inundation area increased slightly as the roughness increased during the rainstorm and
decreased with an increase in the n value during the levee breach when the river flooding was



the main force. These results demonstrate the sensitivity of the model to the roughness.

**4.    Conclusion and discussion**

This study used a simple 2D hydro-inundation model (Floodmap) to investigate serious
compound levee breach-induced flooding during the typhoon "Fitow". The surface runoff
caused by the rainstorm and river overflow were considered in the model. The following
conclusions can be drawn from the simulation results. First, one key advantage of this modeling
approach is the analysis of a single historic flood event. The flooding results showed the time
series of the flooding extent and inundation depth, indicating that the farmland areas near the
river had a very high flood risk. Pluvial flooding or fluvial flooding caused extensive damage
to low-lying areas due to the lack of a drainage network, especially waterlogging of farmland.
The maximum water depth was more than 2 m. Second, within 1-3 h after the dike failure, the
floodwaters spread rapidly, and the inundation area and average water depth reached the peak
value; thus, this is the key period for repairing the levee. Subsequently, the flood risk decreased
as the water level dropped. However, the water does not drain rapidly only by infiltration or
evaporation, and the waterlogging lasted for more than 12 h, resulting in loss of farmland with
high vulnerability. Therefore, in addition to repairing the levee, it is necessary to remove the
flood water in time using drainage measures, such as water pumps.

Model validation was a challenging aspect of this research. The topographic data resolution,
land use, and land cover affect the simulation results. The validation data consisted of field




observations, and the uncertainty associated with incorrect recollections of the residents led to
errors. It was assumed that the error ranged from 5 cm to 15 cm for different land uses. Most
of the verification results matched the field observations and fell within the confidence band,
demonstrating the model's reliability. Nevertheless, some of the simulated water depths were
slightly smaller than the field observations, which was attributed to the exaggeration of the
depth by the respondents.

Another important component of this study is the comparison of the predictions (flooding extent
and average water depth) using different Manning's n values (from 0.01 to 0.1 at a 0.01 interval).
The results demonstrated the model's strong sensitivity to roughness. Overall, the model
exhibited good reliability for single and compound flood modeling. Future research on this
topic should be improved for the following aspects to improve the model robustness. (1)
Higher-resolution topography and hydrological boundary conditions should be used to
represent typical flood conditions. (2) The drainage capacity could be modeled to provide a
more reliable result. (3) Urban compound flood risks should be evaluated to help decision-
makers develop effective emergency response plans and flood adaptation strategies.

**Data Availability Statement**. The raw and processed data from the co-authors' research
findings cannot be shared at this time, as these data are also part of the ongoing research. The
satellite remote sensing image came from the Google Earth open-source datasets
(https://earth.google.com/);

**Author contributions.** YY and JY initiated and led this research. YY designed the flood event
process, analyzed the performance of this model, and wrote the paper. JY provided history
records of water level. WZ and JY gave the suggestion for this paper. YL dealt with the rainfall
data. YZ, AX, YW and WS helped in collecting validation data.





**Acknowledgments**
This paper was supported by the National Natural Science Foundation of China (Grant no:
51761135024, 41871164), the National Key Research and Development Program of China
(Grant no: 2017YFE0107400) and the Shanghai Sailing Program (Grant No. 21YF1456900).

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

crowd-sourced data. Environmental Research Letters, 11(12).





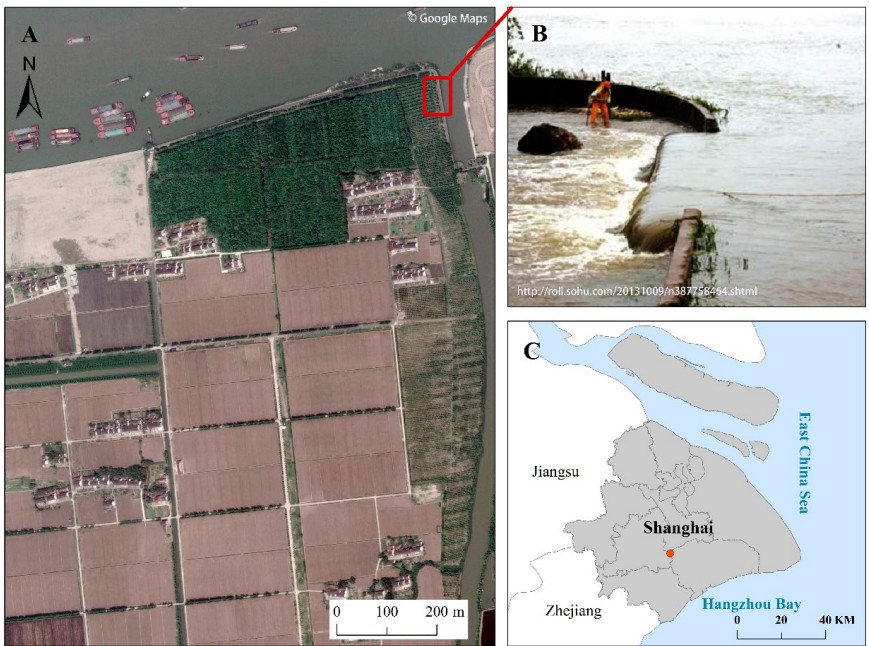


Fig. 1 Location of the study area and levee breach during typhoon "Fitow"





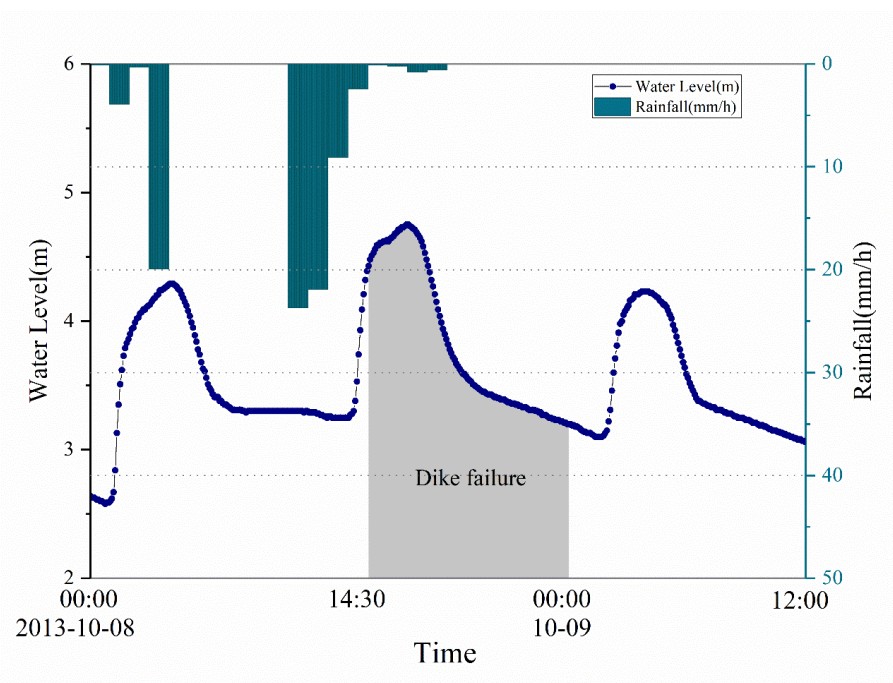


Fig.2 Time series of the water level and rainfall data at Qianbujing Creek during Typhoon

"Fitow"





Levee breach-induced compound flood modeling

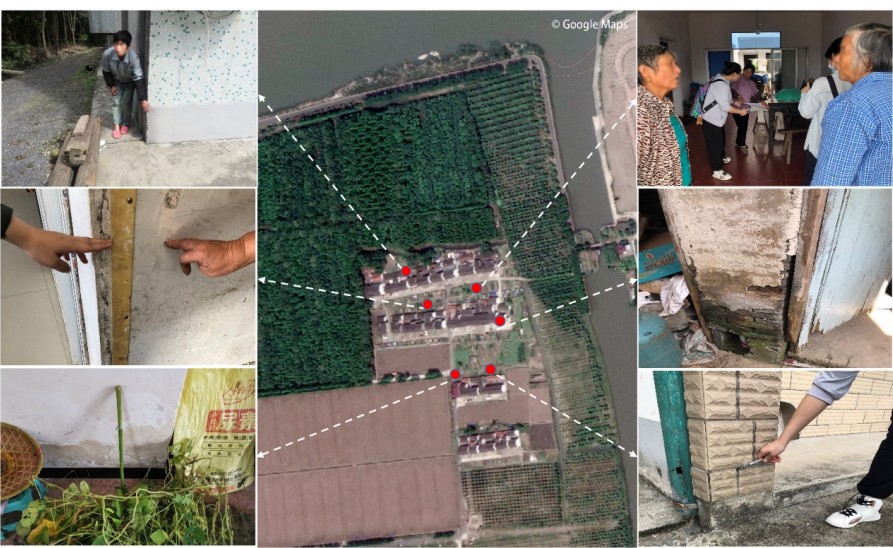


Fig. 3 Field investigation of flood inundation after the event





Levee breach-induced compound flood modeling

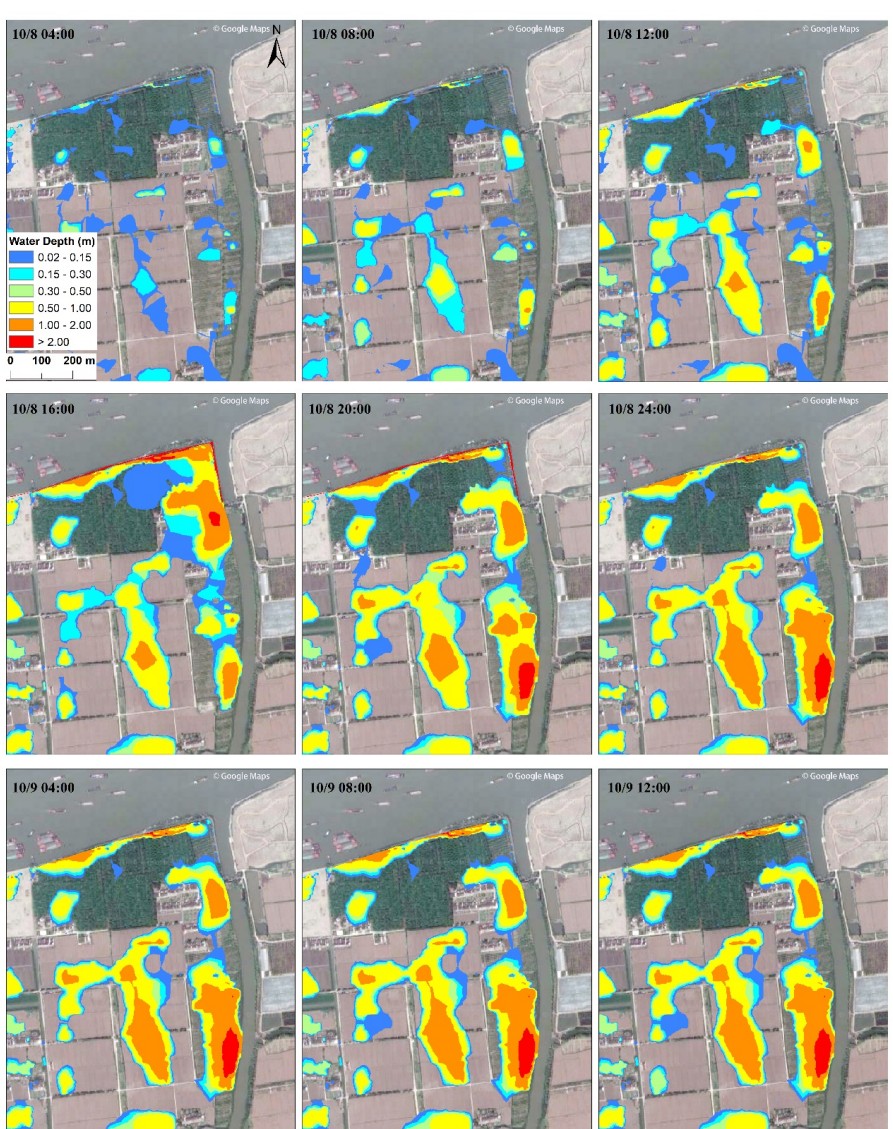


Fig. 4 Time series of flood inundation during the typhoon event



Levee breach-induced compound flood modeling

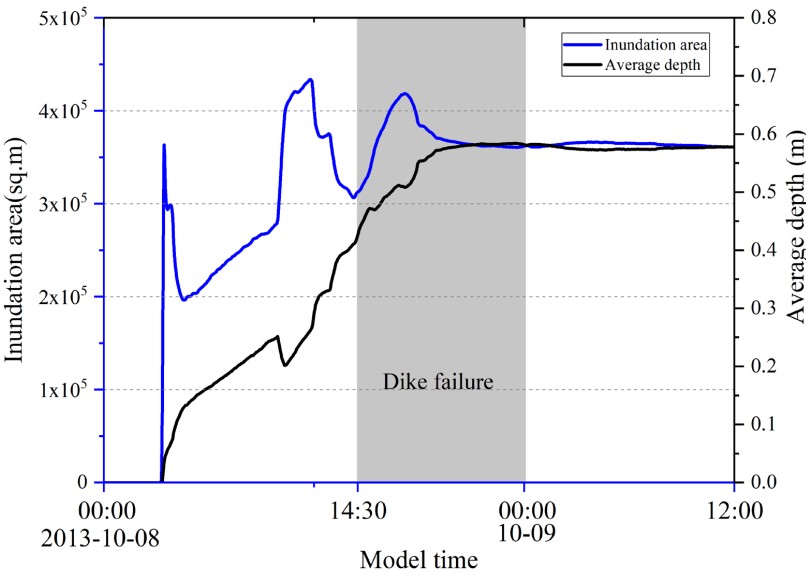


Fig. 5 Time series of the inundation area and water depth during the flood event


Levee breach-induced compound flood modeling

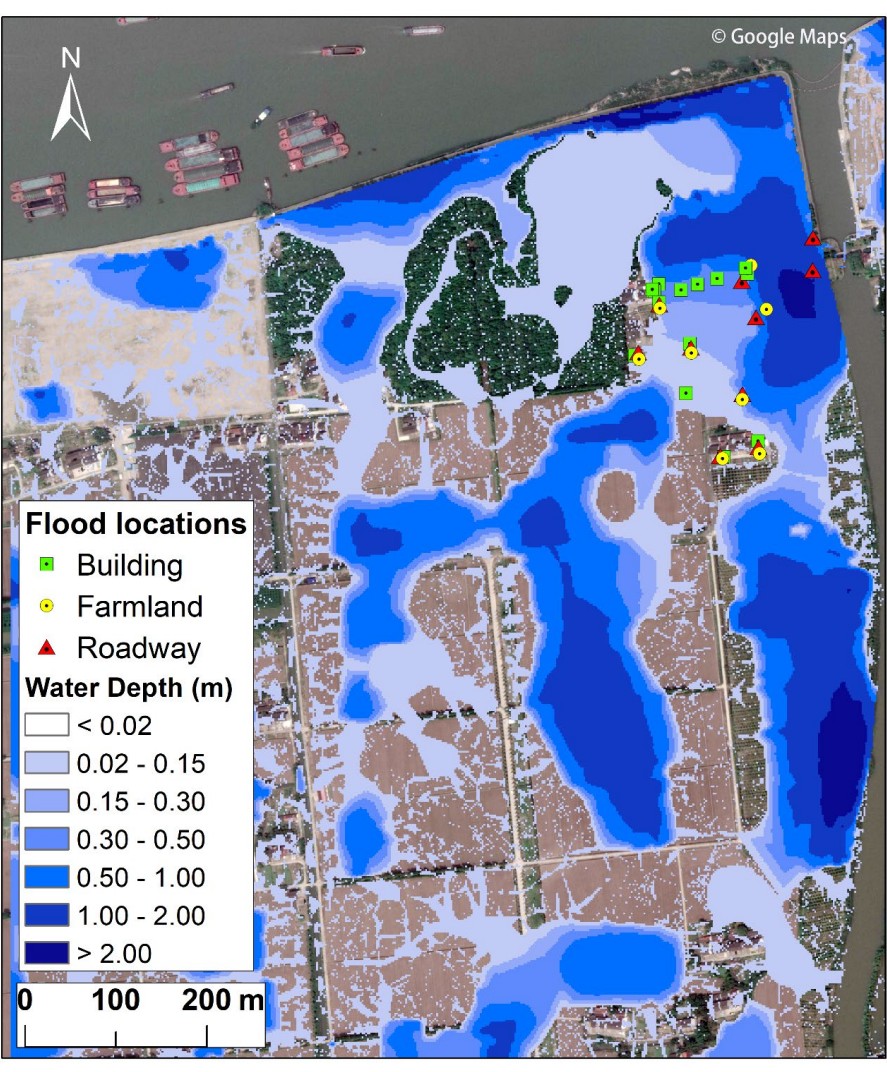


Fig. 6 Maximum flood extent and depth predicted by the model





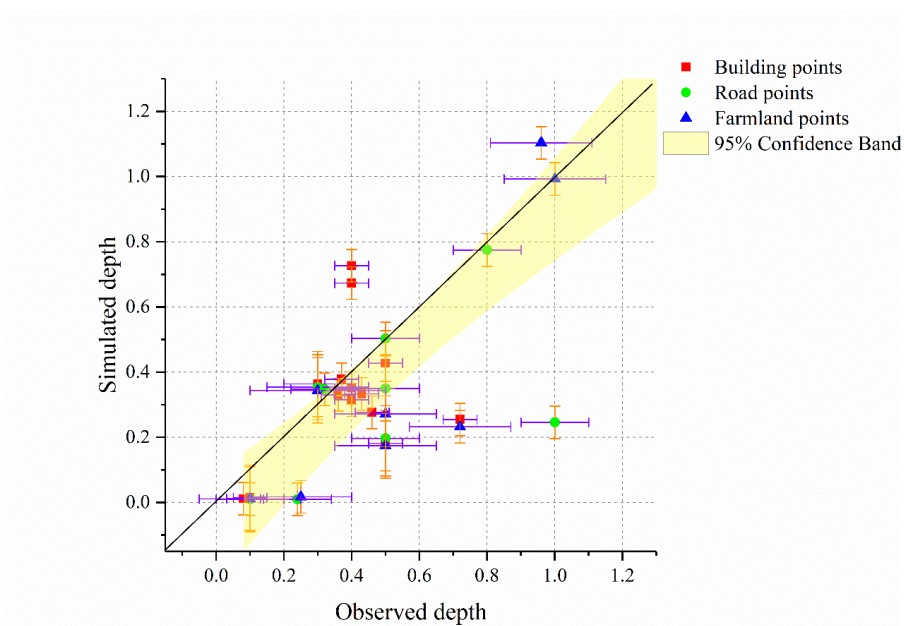

Fig. 7 A comparison of the simulated and observed depths



Levee breach-induced compound flood modeling

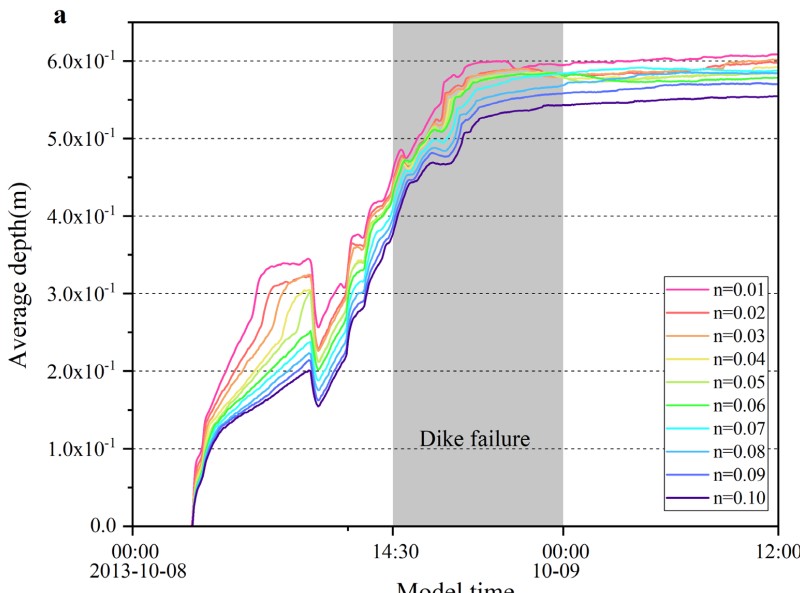

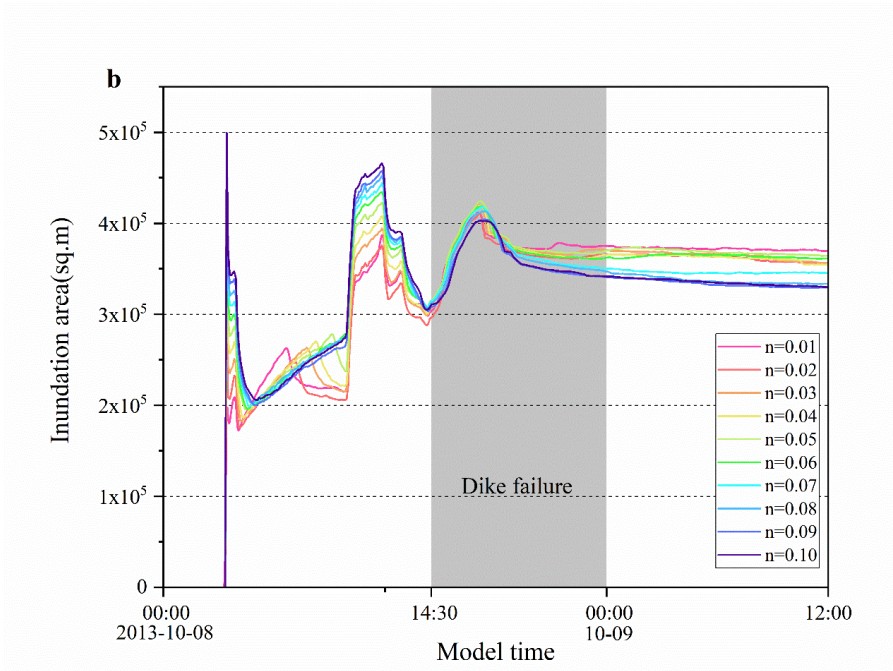

Fig. 8 Sensitivity analysis of the model to Manning's roughness coefficient