# Peer review of "during Typhoon "Fitow" 2 3 Yuhan Yang1, Jie Yin2·3·4\*, Weiguo Zhang1, Yan Zhang2, Yi Lu2, Aoyue Xiao2, Yunxiao Wang2, 4 5 Wenming Song2 6 7 1 State Key Laboratory of Estuarine and Coastal Research, East China No"

_Natural Hazards and Earth System Sciences, 2021_

## Author Comment (AC1)

We sincerely thank Referee #1 for his/her careful review and constructive feedback and suggestions. We truly believe that the changes suggested by Referee #1 will enhance the quality of the manuscript. A point-by-point response is presented below.

**1. what is the innovation of this paper?**

Thanks for your comments, this paper focused on simulating the whole flooding evolution process of a serious levee breach-induced compound flooding by using a 2D hydro-inundation model. On the basis of the historical flooding event, this work revealed the compound effects of levee breach-induced fluvial flooding and heavy rain.

The main innovative aspects are

- It's the first time to investigate the dynamic compound flooding process and mechanism of heavy rain and levee breach-induced flooding.

- Real-life cases of historical flooding events have been adequately investigated which can demonstrate the feasibility and robustness of the model.

**2. It was mentioned that the critical time to minimize damage (should further take actions) is the first a few hours after levee breach; however, readers would perhaps anticipate this conclusion before reading this article.**

Thanks for noting this. We obtained this conclusion from the simulating time-series of spreading flooding scenario, and from the figure below, we can find that the inundation area and the water depth continue to increase rapidly in the early 1-3 h after levee breaching, chiefly because of the water level increasing at the same time, however, during the falling tide period, the flooding diffusion tend to be slow.

We have added this discussion in **Abstract** and **Conclusion** as below:

"Second, within 1-3 h after the dike failure, the floodwaters spread rapidly, and the inundation area and average water depth reached the peak value; chiefly because of the water level increasing at the same time, however, during the falling tide period, the flooding diffusion tend to be slow. Thus, this is the key period for repairing the levee."

[Figure]

**3. In addition, I'm not fully convinced by one of conclusion, which concludes that the model**
**is strongly sensitive to roughness value, maybe it should further explain that why the flood**
**extent not consistently increases with the increasing of the roughness value after levee**
**breach (compares to the flood extent before levee breach)**
Thanks for your suggestion, the sensitivity analysis result statement are inaccurate and we have
revised the description and added the further explain in **Results** as below:
"Interestingly, there were differences in the sensitivity to the roughness before and after the
levee breach for the flood inundation extent. The inundation area increased obviously as the
roughness increased during the rainstorm, however, it is decreased slightly with an increase in
the n value during the levee breach when the river flooding was the main force. The main
reason which causing the sensitivity differences is the unlike formation mechanism of
inundation extent between rainstorm and fluvial flooding."
**Minor comments:**
**4. I feel parts of the introduction is concatenated with literatures (e.g. line 68 to 86), it would**
**be nice to summarize the findings rather than simply list the findings one after another.**
Thank you very much for your suggestion, the Introduction has been changed as below:

A number of approaches for levee breach-induced flood modeling were developed. Some previous studies have investigated the breach mechanism and the hydrological process of dike failure flooding, Vorogushyn (2010) proposed an Inundation Hazard Assessment Model (IHAM), which coupled a 1D hydrodynamic model of river channel routing, a probabilistic dike breach model, and a 2D raster-based inundation model. Cannata et al. (2011) used a GIS-based approach to simplify a 2D dam break simulation. Recent advances have been made in the application of methodologies for predicting the dike failure -induced flooding, Yin et al. (2020) predicted dike failures and flood inundations in Shanghai, China, under various emission scenarios using an interdisciplinary process-based approach.

Similarly, numerous studies analyzed the compound effects of various flood hazards at different scales (Ganguli et al., 2020). Most previous study focus on calculating the joint flood risk probability. For instance, Lian et al (2013) evaluated the joint probability of rainfall and tidal level both exceeding their threshold values through the copula and then analysis the combined effect of them on flood risk in a complex river network in a coastal city in China. Moftakhari et al (2017) proposed a bivariate flood hazard assessment approach to account for compound flooding from river flow and coastal water level. Bevacqua et al (2019) predicted the increasing probability of compound flooding from precipitation and storm surges in Europe under anthropogenic climate change. At a global scale, Couasnon (2020) and Eilander (2020) explored the compound flood potential resulting from storm surges and riverine floods.

5. **Line 25-26: 'In low-elevation areas, temporary drainage measures and flood defenses are equally important'. This has neither studied and nor proved in the paper.**

Thank you very much for your comment, we have added explanation in discussion as below:

"However, the water does not drain rapidly only by infiltration or evaporation especially in low-elevation areas such as farmland, some waterlogging even lasted for more than 12 h (Figure 4), resulting in loss of farmland with high vulnerability. Therefore, in addition to repairing the levee, it is necessary to remove the flood water in time using drainage measures, such as water pumps."

6. **Line 66: what is the economic damage in this area?**

Thanks for your comments, sorry, we cannot find any official reports of the economic damage of this flooding events, and from the field investigation we knew that the government did not assess the property losses of local residents.

7. **How does the model control the levee height during the breaching process in section 2.3?**

Thanks for your comments, we first overlay the remaining intact levee height onto the original bare-earth DEM (remove the 15m levee breach), due to the model cannot change the topography boundary during the running time, so we control the levee height by changing the relative water level, namely before the levee breach, the relative water level is 0 because of there was no flooding, while during the levee breaching period, the relative water level is the historical river water level, so that the flood spread from the breach section.

8.  **Line 184: what does 'remaining' mean? Does it mean the height of floodwall was decreased to 4.9-5m?**

Thanks for your comments, the 'remaining' means the remaining intact floodwall without the breach section, and the levee height was about 5 m above Wusong Datum.

We have added the description in section 2.3 in more detail:

"The leveel height and location were obtained from the Shanghai Municipal Institute of Surveying and Mapping. The height of remaining intact floodwall without the breach section (about 5 m above Wusong Datum) were then overlaid onto the original bare-earth DEM using the raster calculator in ArcGIS 10.6 software."

9.  **Line 269: the assumed evapotranspiration value is based on what?**

Thanks for your comments, and we have added description about evapotranspiration value

"we assumed evapotranspiration of 3 mm/day, a value that which generates a good inundation prediction in the urbanized area (Yin et al., 2016; Yu and Coulthard, 2015)."

10. **it would be better to combine figure 2 and 5, which can clearly show the inundation process due to rainfall and sustained high water level in the river.**

Thanks for your suggestion, the figure 5 have been changed as below:

[Figure]

**11. Line 321: maybe it's better to show the breach location or highlight waterfront area in Figure 4.**

Thanks for your suggestion, we have added the breach location in figure 4:

[Figure]

**12. Line 348: decreased from 0.6 to 0.55?**

Thanks for noting this. It has been corrected.

**13. In Figure 3, it seems that these six points are building locations, how about the roadway and farmland? This is not consistent to the text line 367-369.**

Thanks for your suggestion, in Figure 3, we choose some representative sample of flood locations, and the water depth marks of these points are relatively clear. However, the water depth records of roads or farmland were all from investigates' dictation and lots of these locations were repaired after that event, so we didn't take too many reliable pictures.

---

## Author Comment (AC2)

We sincerely thank Referee #2 for his/her careful review and valuable advices. Based on the comments and suggestion, we read through comments carefully and have made extensive modification on the original manuscript. The responses to the reviewer's comments are marked in blue and presented following.

1. **In abstract, there shall be a statement on the effectiveness of the 2D Floodmap model. To what extent do the modeling results match with the real flood patterns? What are the advantages of the model? This is a key message that must be clearly stated in the abstract.**

   Thanks for your comments, we have added the description of Floodmap in the Abstract as below:

   "…using a well-established 2D hydro-inundation model (Floodmap) to reconstruct this typical event. This model couples urban hydrological processes with flood inundation for high-resolution flood modeling, which has been applied in a number of different environments and now Floodmap is the mainstream numerical simulation model used for flood scenarios."

2. **Page 3, line 54-56, the "Himalaya glacier outburst flood in northern India" needs a reference**

   Thanks for your comments, the sentence was reference from the news (Scores missing as Himalayan glacier bursts in northern India (france24.com)),we have added the reference in paper.

3. **Figure 1, the sources of the image and picture shall be clarified.**

   Thanks for your comments, Figure 1(A) was from Google Maps, and the Figure 1(B) was from historical news, we have added the picture source URL and the Copyright information on each picture.

[Figure]

4. **Figure 2, there needs more explanations of the three peaks of the water level curve. Especially why is there a "third peak", what caused it?**

Thanks for your question, Figure 2 shows 36 h riverine tidal hydrographs of Huangpu River, the three peaks were main due to the rising tidal process but not the rainfall. However, the heavy rain also directly increased the water level (the second peak).

5. **As I understand, Compound flooding is an extreme impact event resulting from the interaction of multiple drivers (Zscheischler et al. 2018), mostly rainfall and tides (Bavacqua, et al 2020). But in this study, the flood was obviously mainly caused by heavy rainfall from Fitow, and there seems no tides or storm surges at the study site. Thus I suppose the authors shall not call it as a compound flood.**

Thanks for your comments. It's a very important question, in previous study, compound flooding mostly due to the co-occurrence of high sea level and precipitation, however, in this study, the levee breach was caused by the compound effect of the rapid rising riverine water level and the heavy rain. The flood not only caused by the heavy rain but also the record-breaking river tides by the storm surge brought by the Typhoon "Fitow". When the heavy rain met the rising riverine tides, what made water level much higher, resulting in levee breach. Therefore, this flooding event was considered as a compound flooding in this paper.

6. **The discussion and conclusion section is weak. The discussion shall be improved and extended with more on the possible strategies and measures to reduce such levee breach and associated risks. E.g. according to the flood pattern and process, which areas and which measures could be most effective in reducing the flood impacts? From engineering perspective, how could the levee be strengthened, to which level? In addition, it would be also valuable to compare the present study findings with other Some flood adaptation studies and household responses measures may be referred and compared, for instance:**

Thank you very much for your suggestion. The discussion regarding this question is presented following:
"…. Third, the water does not drain rapidly only by infiltration or evaporation, and the waterlogging lasted for more than 12 h, resulting in loss of farmland with high vulnerability. Therefore, for levee breach-induced flood response in rural area, in addition to repairing the levee in time, it is necessary to remove the flood water using drainage measures at the same time, such as setting water pumps near the farmland or other low-lying area, when necessary, government should guide nearby residents to evacuate to a safe place as well.

Beyond the flood emergency response measures, effective long-term engineering measures may be more suitable for fundamentally decreasing the unpredictable levee-breach flooding risk, local specifications for the flood-control engineering should be updated with the increasing flood risk in the context of climate change (Yang et al., 2015)."

**7. Meanwhile, there is no conclusion in the current section 4. I would suggest to add a paragraph to summarize the key findings in this study, with simple and clear sentences. This helps readers to quickly get the key points of the study.**

We deeply appreciate your suggestion. We have added some sentences to summarize the findings in **section 4** as below:

"Simulation of real-life historical severe flooding events can reveal the dynamic flooding process and mechanism. In this study, a serious compound levee breach-induced flooding during the typhoon "Fitow" have been adequately investigated used by a simple 2D hydro-inundation model (Floodmap). The surface runoff caused by the rainstorm and river overflow were considered well in the model.

The following conclusions can be drawn from the simulation results. First, one key advantage of this modeling approach is the analysis of a single historic flood event. The flooding results showed the time series of the flooding extent and inundation depth, indicating that the low-lying area especially for farmland areas near the river had a very high flood risk. The compound flooding caused extensive damage to low-lying areas not only due to the elevation but the lack of a drainage network, resulting in the average water depth over 0.5 m more than 12 h. Second, within 1-3 h after the dike failure, the floodwaters spread rapidly, and the inundation area and average water depth reached the peak value, chiefly because of the rising riverine tides at the same time, however, during the falling tide period, although the dike has not been repaired, the flooding diffusion tend to be slow, the flood risk decreased as the water level dropped as well. Thus, it can be indicated that the levee breach-induced flooding spread was heavily dependent on the change of riverine tides, the key period for levee breach-induced flooding control (such as repairing the levee, evacuation) was from levee breach to the end of rising tide. Third, the water does not drain rapidly only by infiltration or evaporation, and the waterlogging lasted for more than 12 h, resulting in loss of farmland with high vulnerability. Therefore, for levee breach-induced flood response in rural area, in addition to repairing the levee in time, it is necessary to remove the flood water using drainage measures at the same time, such as setting water pumps near the farmland or other low-lying area, when necessary, government should guide nearby residents to evacuate to a safe place as well."

[Figure]

8. **The English language is in general not sufficiently good for a scientific publication, which must be further modified.**

   Thanks for your comments, we apologize for the language problems in the original manuscript. The paper will be carefully revised to improve the grammar and readability.

---

## Author Response (AR2)

**Response to comments**

We thank all referees and editors for their important advices. Based on the new comments, we make a revision, hope it will improve our manuscript. A point-by-point responses to the comments are marked in blue and presented following.

Referee 1

1. **The innovation of this paper still shows somehow weakness. The methods and modelling work seem sound but it's still not clear what this paper would like to convey in a scientific sense; in addition, it is not clear what research questions have been answered. It's not a simply first time of study on levee breach flooding and rainstorm induced flooding (many papers have studied on combined consequence of the fluvial and pluvial flooding). The levee breach flooding is caused by high water level in the river in this case. The authors should examine the high water level is contributed by high runoff from upstream or high tide level coming from the estuary, or the combination. I think the 'introduction' part should be largely improved.**

Thank you very much for your comments. This manuscript mainly focused on simulating and verifying the hydrodynamic process of the levee breach-induced flooding and the rainfall-runoff. Although many previous articles have paid attention to the occurrence probability and impact of levee breach flooding and rainstorm-induced flooding, there are few articles that verify the results based on historical events. Therefore, to deeper digging the dynamic multiple flooding processes, in this manuscript, a real-life case of historical flooding events has been adequately investigated. Our results not only provide a comprehensive view of the spatial patterns of the flood evolution but also verifies the model.

We have stressed more the novelty of this paper in the Introduction (**Line 59**)

"In addition, the co-occurrence or subsequent occurrence of multiple flood drivers such as coastal high tide, storm surge, extreme precipitation, and high river flow resulting in large runoff may cause compound flooding. The compound effect is much greater than the effect of individual flood events (Wahl et al., 2015; Ghanbari et al. 2021). For instance, typhoon Fitow in 2013 brought torrential rain and caused high storm surges, resulting in record-breaking riverine water levels in the upstream region of the Huangpu River, Shanghai, China. As a result, the floodwall along the upstream Qianbujing Creek could not withstand the high water level, leading to a breach in a 15-m long section at 14:30 on 8 Oct 2013. Although the broken section was repaired after about 8 hours, the levee breach combined with heavy precipitation resulted in extensive flood inundation in the rural areas.
......
The above studies contributed significantly to the modeling and evaluation of dike failure-induced flooding, as well as compound flood risk. However, most previous studies have paid attention to the occurrence probability and final impact of compound flooding, but few of them investigated the complete compound dynamic hydrological process of these extreme cases. Moreover, historical compound flooding events were not adequately investigated in previous

articles, these real-life cases play an important role to demonstrate the feasibility and robustness of study results. To address the research gaps, this case study seeks to examine the changing nature of levee breach-induced compound flooding. A 2D hydro-inundation model Floodmap is used to simulate the process of the compound flood event that occurred in Qianbujing Creek to improve our understanding of the evolution of flood inundation. The results of the approach are validated by field measurements, including the inundation depth and the flood extent over time. The findings can provide support for decision-makers to develop flood adaptation measures."

2. **I suggest not mention compound effects, use 'dynamic multiple (or combined) flooding processes' instead. The compound flooding process is caused by the interaction of multiple physical or human-being induced drivers. While in this paper the levee breach flooding is the same cause of heavy rain which causes high runoff (and water stage) in the river during typhoon.**

Thanks for your suggestion, compound flooding refers to a phenomenon in which two or more flooding sources occur simultaneously or subsequently within a short period of time. actually, the compound flooding process mentioned in our manuscript were caused by high precipitation, and high river flow. Typhoon Fitow in 2013 brought torrential rain and caused high storm surges, resulting in record-breaking riverine water levels in the upstream region of the Huangpu River, Shanghai, China (Line 62). So that the interaction among these flood drivers caused a compound flood event.

3. **refer to the answers to the major comment (3), what are the differences of formation mechanism of inundation in rainstorm flooding and fluvial flooding separately? please explain it in details in the text.**

Thanks for noting us, we have added the further explain in the text (**Line 359**)

Interestingly, there are differences in the sensitivity to the roughness before and after the levee breach for the flood inundation extent. The inundation area increases as the roughness rise during the rainstorm. However, the inundation area decreases slightly with the growth of the n value during the levee breach when the river flow is the major cause of the flooding. As a result, the rainfall is more likely to cause ponding with high roughness, as it drops the flow velocity. Whereas, when the river flow is the main force, the decline of roughness value leads to an increase in flood velocity which accelerates the spread of flood. These results demonstrate the sensitivity of the model to the roughness.

4. **Refer to the answers to the minor comment (7), there is an assumption of sudden collapse of levee breach during the breaching process. I suggest the authors should mention it and explain it in the section 2.3.**

Thanks for noting us, we have added the explain in the section 2.3.(**Line 191**)

"Due to the model cannot change the topography boundary during the running time, so we control the levee height by changing the relative water level, namely before the levee breach, the relative water level is 0 because there was no flooding, while during the levee breaching period, the relative water level is the historical river water level, so that the flood spread from the breach section."

Referee 2

5.   I still have several comments and would suggest the authors to make a minor revision. Title of the paper is not well worded. I would suggest the authors to consider "Modeling of a compound flood induced by the levee breach at Qianbujing Creek, Shanghai during Typhoon Fitow".
I don't think it is necessary to use quotation marks for Fitow.

Thank you very much for your suggestion, the title has been changed as below:
 "Modeling of a compound flood induced by the levee breach at Qianbujing Creek, Shanghai during Typhoon Fitow"

6.   In 2.2.1, a DSM of 6m resolution is the original data source of elevation of the study area. I suppose the 6m is horizontal resolution. Then how is the resolution at vertical direction? I mean if your elevation data has only one-meter vertical resolution, you are not able to model the precise inundation less than 1m.
Thanks for noting us, the vertical resolution was 0.1-0.2m, we have added the description as below (**Line 127**):
"we use a high resolution digital surface model (DSM) with 6-m horizontal resolution, 0.1-0.2m vertical resolution"

7.   **Figure 2 curves are repeated in figure 5. I think you can delete figure 2, and you better improve figure 5. You may consider re-edit figure 5 similar to the style and layout of the "Figure 4 in Zhang, et al., 2011" https://www.pnas.org/content/108/42/17296**

Thank you very much for your suggestion, we have improved figure 5 and deleted figure 2 (**Line 528**)

We also revised the format of figure 6,7a,7b, please find them in Supplement

8.  **Section 4, the heading shall be "Discussion and conclusion". It is often discussion first then make conclusions.**

    Thanks for your comments, we have changed the structure of manuscript according to the content, section 3 is changed as "Results and Discussion" and section 4 is changed as "Conclusion"

9.  **Still the language is not of good quality, with many uncommon wording and sentence structures. Some sentences are very long. It is certainly necessary to further improve this.**

    We feel sorry for the language problems in manuscript. We have revised the whole manuscript with the assistance from a colleague whose English is good. The revised version should be more readable.